# Liraglutide Suppresses Tau Hyperphosphorylation, Amyloid Beta Accumulation through Regulating Neuronal Insulin Signaling and BACE-1 Activity

**DOI:** 10.3390/ijms21051725

**Published:** 2020-03-03

**Authors:** Salinee Jantrapirom, Wutigri Nimlamool, Nipon Chattipakorn, Siriporn Chattipakorn, Piya Temviriyanukul, Woorawee Inthachat, Piyarat Govitrapong, Saranyapin Potikanond

**Affiliations:** 1Department of Pharmacology, Faculty of Medicine, Chiang Mai University, Chiang Mai 50200, Thailand; salinee.jan@cmu.ac.th (S.J.); wutigri.nimlamool@cmu.ac.th (W.N.); 2Research Center of Pharmaceutical Nanotechnology, Chiang Mai University, Chiang Mai 50200, Thailand; 3Neurophysiology Unit, Cardiac Electrophysiology Research and Training Center, Faculty of Medicine, Chiang Mai University, Chiang Mai 50200, Thailand; nipon.chat@cmu.ac.th (N.C.); siriporn.c@cmu.ac.th (S.C.); 4Department of Physiology, Faculty of Medicine, Chiang Mai University, Chiang Mai 50200, Thailand; 5Department of Oral Biology and Diagnostic Sciences, Faculty of Dentistry, Chiang Mai University, Chiang Mai 50200, Thailand; 6Institute of Nutrition, Mahidol University, Salaya, Phuttamonthon, Nakhon Pathom 73170, Thailandwoorawee.int@mahidol.ac.th (W.I.); 7Department of Pharmacology, Faculty of Science, Mahidol University, Bangkok 10400, Thailand; piyarat@cgi.ac.th; 8Chulabhorn Graduate Institute, Chulabhorn Royal Academy, Bangkok 10210, Thailand

**Keywords:** liraglutide, Alzheimer’s disease, SH-SY5Y, tau, beta-amyloid, insulin resistance, BACE-1

## Abstract

Neuronal insulin resistance is a significant feature of Alzheimer’s disease (AD). Accumulated evidence has revealed the possible neuroprotective mechanisms of antidiabetic drugs in AD. Liraglutide, a glucagon-like peptide-1 (GLP-1) analog and an antidiabetic agent, has a benefit in improving a peripheral insulin resistance. However, the neuronal effect of liraglutide on the model of neuronal insulin resistance with Alzheimer’s formation has not been thoroughly investigated. The present study discovered that liraglutide alleviated neuronal insulin resistance and reduced beta-amyloid formation and tau hyperphosphorylation in a human neuroblostoma cell line, SH-SY5Y. Liraglutide could effectively reverse deleterious effects of insulin overstimulation. In particular, the drug reversed the phosphorylation status of insulin receptors and its major downstream signaling molecules including insulin receptor substrate 1 (IRS-1), protein kinase B (AKT), and glycogen synthase kinase 3 beta (GSK-3β). Moreover, liraglutide reduced the activity of beta secretase 1 (BACE-1) enzyme, which then decreased the formation of beta-amyloid in insulin-resistant cells. This indicated that liraglutide can reverse the defect of phosphorylation status of insulin signal transduction but also inhibit the formation of pathogenic Alzheimer’s proteins like Aβ in neuronal cells. We herein provided the possibility that the liraglutide-based therapy may be able to reduce such deleterious effects caused by insulin resistance. In view of the beneficial effects of liraglutide administration, these findings suggest that the use of liraglutide may be a promising therapy for AD with insulin-resistant condition.

## 1. Introduction

Insulin resistance is defined as an inefficient response of tissue to normal plasma insulin levels [1]. Previous studies have shown that insulin plays an important role in several brain functions, including neuronal synaptic plasticity and cognitive function [2,3]. Accumulated evidence has revealed that an impaired insulin sensitivity is positively correlated with a development of Alzheimer’s disease (AD)-related pathology [4,5,6,7,8,9,10,11]. For example, (1) insulin receptors (IRs) in the brain have been shown to be downregulated in AD models [8]; (2) the impairment of insulin signaling has been documented in the brain from both postmortem analysis of AD patients and in animal models of AD [6,7,8,9]; and (3) the deficit of memory function in AD patients has been shown to be associated with altered levels of circulating insulin, insulin levels in the brain, and defects in peripheral insulin signaling pathways [4,5,10]. It is well known that insulin signaling is associated with the regulation of tau protein, and deregulation of brain insulin signaling is linked to AD [12]. For example, systemic insulin resistance results in tau hyperphosphorylation detected in cerebrospinal fluid [13]. Hyperphosphorylated tau can eventually lead to neurofibrillary tangle formation [14,15]. It has been reported that insulin signaling accelerates amyloid-beta trafficking from the trans-Golgi network to the plasma membrane and reduces the intracellular accumulation of beta-amyloid [16]. Therefore, a defect in insulin pathway results in Aβ accumulation. This statement is supported by a study reporting that induction of insulin resistance in Tg2576 mice reduced the amount and activity of insulin-degrading enzyme (IDE), contributing to an increase in Aβ levels in the hippocampus and cerebral cortex [17]. The defined mechanism of tau phosphorylation has been reported to be due to the glycogen synthase kinase 3 beta (GSK-3β), a Tau kinase regulated by insulin via the protein kinase B (AKT) pathway [18]. Furthermore, the newly identified category of AD as type 3 diabetes [11] has encouraged researchers to focus on antidiabetic drugs as a new strategy for treating AD associated with insulin resistance.

A previous study in mouse neuroblastoma cells showed that chronic hyperinsulinemic conditions induced neuronal insulin resistance as well as the development of molecular markers of AD, including amyloid-β and tau hyperphosphorylation [19]. The study demonstrated that a widely used antidiabetic drug, metformin, ameliorated neuronal insulin resistance and those AD markers [19]. Although metformin usually has minimal adverse effects, its rare but serious side effect of lactic acidosis is a safety concern in some cases [20]. Besides metformin, which is a standard drug for diabetes mellitus type 2 (T2DM), there has been an attempt to explore the efficacy of some natural-derived compounds. For example, silybin a compound shown to exert a protective effect against diabetes and its complication [21], has been recently demonstrated to possess crucial anti-AD properties [22]. Incretin-based therapy, and newly developed insulin sensitizers, including glucagon-like peptide 1(GLP-1) agonists, are widely used to treat T2DM [23,24,25]. Those sensitizers have many favorable metabolic profiles, including weight loss, and reduce the risk for hypoglycemia. Moreover, recent studies of GLP-1 receptor agonist therapy in T2DM have been shown to improve cognitive function, including learning and memory [26,27,28,29]. However, effects of GLP-1 receptor agonists, particularly liraglutide, on neuronal insulin resistance., as well as AD-associated neuropathological characteristics in human neuroblastoma cell lines, have not yet been clearly investigated. The present study aimed to characterize the molecular signaling underlying neuronal insulin-resistance-induced Alzheimer-like changes on hyperinsulinemic human neuroblostoma cell line, SH-SY5Y, as well as to determine the therapeutic effect of liraglutide on those pathological conditions. Here we reported that liraglutide could be able to significantly ameliorate the development of Alzheimer-like changes in the insulin-resistant-SH-SY5Y.

## 2. Results

### 2.1. Hyperinsulinemic Condition Induced Neuronal Insulin Resistance in a Human Neuroblastoma Cell Line, SH-SY5Y

The physiological concentration of human plasma insulin is approximately 1 nM [30]. In the present study, a high concentration of insulin (100 nM) was used to induce neuronal insulin resistance without exerted cytotoxicity to the cells (Appendix A) [31]. Following a 48 h hyperinsulinemic condition, neuronal insulin signaling molecules, including insulin receptor (IR), insulin receptor substrate 1 (IRS-1), AKT/PKB, and GSK-3β and their phosphorylation status were observed (Figure 1). The control cells showed a normal response to insulin stimulation observed by the phosphorylations of IR (Tyr1162/1163), pIRS-1(Tyr)/IRS-1, and AKT/PKB (Ser473), while the insulin resistant-cells significantly decreased those of phosphorylations (Figure 2A–C compared C+/C to I+/I). Moreover, the GSK-3β activity, which is thought to be inactivated by phosphorylation at serine 9, was increased in the insulin-resistant cells (Figure 2D, comparing C+/C to I+/I). These findings indicated that 48 h treatment of 100 nM insulin administration was enough to induce neuronal insulin resistance.

### 2.2. Hyperinsulinemic Condition Increased the Expression of Apoptotic Protein Markers and the Formation of Alzheimer’s Markers in Neuronal Cells

Previous evidence has shown that the impairment of brain insulin signaling is associated with progressive neurodegenerative diseases such as AD and neuronal death [8,32,33]. Despite the fact that insulin signaling is considered as an important pathway to promote neuronal survival, the hyperinsulinemic condition might lead to an increase in the expression of a pro-apoptotic protein, Bax (Figure 3A, comparing C to I). However, a hyperinsulinemic condition in our study was not strong enough to observe a decrease of an anti-apoptotic protein, Bcl-2 (Figure 3A compared C to I). Interestingly, pathological proteins associated with AD, including Aβ and hyperphosphorylation of tau protein, were significantly increased under the hyperinsulinemic condition (Figure 3B,C, comparing C to I). To confirm the formation of Aβ plaques or neurofibrillary tangles in hyperinsulinemic condition, extracellular proteinaceous insoluble plaques were stained using Thioflavin-S dye in order to detect the fibrillary β-sheet-rich proteins such as Aβ plaques or neurofibrillary tangles. The results showed that a chronic hyperinsulinemic condition significantly increased an extracellular plaque formation when compared with the control group (Figure 3D, comparing C to I).

### 2.3. Liraglutide Restored Neuronal Insulin Sensitivity in Hyperinsulinemic Conditions

Liraglutide at 500 nM, the optimal concentration which exerted no toxicity to neuron was used to determine the rescue effect on a neuronal insulin-resistant condition (Appendix A). The 24 h liraglutide treatment after the hyperinsulinemic induction significantly restored neuronal insulin sensitivity by improving the phosphorylation status of IR, IRS-1, AKT/PKB, and GSK-3β, when compared with the insulin-resistant group (Figure 4A–D, comparing I+/I and IL+/IL).

The combination effects of aging and insulin resistance were found to be associated with a lack of glucose transportation and its metabolism [34]. We therefore tested whether liraglutide can restore glucose uptake in neuronal insulin-resistant cells. Under hyperinsulinemic condition, glucose uptake was clearly decreased compared to the control (Figure 4E, comparing C to I). However, liraglutide treatment could not restore the glucose uptake in insulin-resistant cells even if the insulin signaling was improved (Figure 4E, comparing C to I).

### 2.4. Liraglutide Decreased the Formation of Alzheimer’s Markers and Plaque 

As noted, 48 h insulin application resulted in an increased Bax and decreased Bcl-2 expression (Figure 3A and Figure 5A, compare C and I). However, by 24 h of liraglutide administration, the dysregulation of Bax and Bcl-2 in a hyperinsulinemic condition was not enough to rescue (Figure 5A, compare I and IL). When we look through the expression of AD markers such as β-amyloid proteins and tau phosphorylation, the restoration was markedly observed in insulin-resistant cells after 24 h liraglutide treatment (Figure 5B,C, compare I and IL). Moreover, liraglutide efficiently reduced plaque formation observed by the bright green fluorescent signals of thioflavin-S fluorescent staining (Figure 5D, compare I and IL). These findings suggest that liraglutide plays an important role in the reduction of amyloid plaque in hyperinsulinemic condition, although the 24 h treatment period was not enough to induce the expression of anti-apoptotic protein or to reduce the pro-apoptotic protein.

### 2.5. Liraglutide Inhibited Beta Secretase 1 (BACE-1) Activity in Insulin-Induced Insulin Resistance in Neuronal Cells

Due to the fact that liraglutide treatment in the hyperinsulinemic condition showed rescue insulin signaling defects together with an improvement in the AD markers, the mechanism of improvement might be speculated as (1) the AD markers were reduced mainly because of the rescue in insulin signaling pathway, or (2) it may also be caused by other insulin signaling-independent mechanisms. As we know, there are several enzymes participating in the toxic amyloid-beta generation: the β-secretase, also known as β-site amyloid precursor protein cleaving enzyme or BACE-1, initiates cleaving an extracellular domain of amyloid precursor protein 1 (APP1); then, the APP1 is cleaved by the γ-secretase to generate Aβ peptides. The BACE-1 enzyme is especially well-documented as having a crucial role in AD pathogenesis [35]. Therefore, we would like to know whether the BACE-1 activity is deprived under a hyperinsulinemic condition and whether the liraglutide can improve it. To test the hypothesis, the activity of BACE-1 was performed by the specific BACE-1 assay. The results showed that during a hyperinsulinemic condition, the BACE-1 activity was prominently increased and it turned to be normal after liraglutide treatment (Figure 4F). Cumulatively, the AD marker ameliorations in our study were possibly mediated by both insulin signaling as well as BACE-1 activity improvement.

## 3. Discussion

Since neuronal insulin resistance and hyperinsulinemia are considered as the important risk factors in cognitive impairment as well as AD, several antidiabetic drugs like insulin sensitizers have been currently investigated for the treatment of this disease [36,37]. Recently, a novel GLP-1 analog called liraglutide showed a promising result in a preventive declination of brain glucose metabolism [38]. So far, a suitable model to study the effect of liraglutide in insulin-resistant and hyperinsulinemic conditions in neuronal cells has not been clearly developed. We therefore developed the feasible model of neuronal insulin resistance under a hyperinsulinemic condition by using the SH-SY5Y, a well-documented cell line which is widely used to study the pathology of AD and its effective drugs. In this study, we tested the effect of liraglutide in an in vitro neuronal insulin resistance model. The major findings are: (1) the 48 h of hyperinsulinemic condition is enough to induce neuronal insulin resistance, increase neuronal apoptotic protein markers and AD markers, and augment the formation of extracellular plaques; (2) the 24 h liraglutide treatment restores neuronal insulin sensitivity and decreases the formation of AD markers and plaque formation, but is not enough to affect any changes in neuronal apoptotic markers and glucose uptake in the hyperinsulinemic condition; and (3) liraglutide suppresses BACE-1 activity which is accelerated under a hyperinsulinemic condition.

Several studies have revealed the important roles of insulin signaling in the normal brain. Its defects could be harmful to neurons and finally lead to the development of neurodegenerative disorders [19,39]. The present study provides evidence supporting the idea that hyperinsulinemic condition causes neuronal insulin resistance as indicated by the decreased phosphorylation of insulin receptors and their downstream signaling molecules, including IRS-1, AKT/PKB, and GSK-3β. AKT usually acts as an inhibitor of GSK by inducing the phosphorylation on N-terminal serine 21 of GSK-3α and serine 9 of GSK-3β, thus diminishing prime-substrate phosphorylation by GSK-3 [19,40]. GSK-3β, a major tau kinase, has been shown to induce an abnormal tau hyperphosphorylation [40,41,42]. Tau hyperphosphorylation results in an instability of microtubules, which causes the formation of neurofibrillary tangles (NFTs), one of the histopathological hallmarks of AD [14,15,43]. The present study demonstrated that tau kinase activity was increased via decreased phosphorylation of GSK-3β at serine 9 residue. Consistently, Gupta and colleagues also demonstrated an increase of tau hyperphosphorylation by GSK-3β in the mouse N2A cell line with a hyperinsulinemic condition [19]. An increase in GSK-3β phosphorylation following hyperinsulinemic conditions might impair Aβ clearance, and it could be one of the reasons to increase Aβ formation in our study. The production of Aβ is generated by BACE-1 or β-secretase, which cleaves an extracellular domain of APP [44]. The previous study revealed that GSK-3β inhibition could reduce BACE-1-mediated cleavage of APP via NF-κB signaling-dependent pathway [45]. Currently, β-secretase inhibition is being considered as one of the strategies to decrease Aβ concentrations, which might in turn prevent or treat AD. In the present study, the hyperinsulinemic condition enhanced the activity of BACE-1 which reflected an accumulation of Aβ protein. Interestingly, liraglutide drastically reduced Aβ production via an improvement of insulin signaling pathway as well as a decrease of BACE-1 activity. The cumulative results indicated that liraglutide has a potential to lower amyloid aggregation and may improve Alzheimer’s symptoms via insulin signaling as well as BACE-1. However, further study is needed to explore the underlying mechanism of how liraglutide affects BACE-1 activity.

The present study also demonstrated that hyperinsulinemic condition or neuronal insulin-resistant condition correlated with an increase expression of a pro-apoptotic protein, Bax, and a decreased expression of an anti-apoptotic protein, Bcl-2. Unfortunately, the 24 h liraglutide was not enough to rescue Bax and Bcl-2 dysregulation in the neuronal insulin-resistant condition. In this case, time and duration of drug administration should be taken into consideration in further study. For instance, the previous study revealed benefits of a pretreatment strategy in cells with an exposure of methylglyoxal stress. Liraglutide applied before the stress exposure showed a decrease in Bax expression [46], while our study used the post-treatment of liraglutide after the appearance of neuronal insulin resistance, which could reveal an importance of time to administration.

Several antidiabetic drugs have been reported to improve insulin sensitivity so far [33,47,48]. GLP-1 agonists are a promising therapy for treating insulin resistance in both peripheral and central compartments [49,50]. GLP-1 has been shown to induce neurite outgrowth and to prevent oxidative injury and cell death [51,52], as well as to improve insulin sensitivity in both T2DM models and AD models [53]. The present study demonstrated the benefits of liraglutide in ameliorating neuronal insulin resistance and AD markers. These findings are also supported by previous studies [53]. For example, (1) Yang and colleagues have shown that liraglutide reversed the impairment of AKT and GSK-3β phosphorylation and also ameliorated tau hyperphosphorylation in type 2 DM rats [53]; and (2) McLean and colleagues also demonstrated that liraglutide reduced Aβ plaque deposition with improving cognitive function in early-stage APP/PS1 mice with AD [54] and in late-stage AD mice [55].

We herein provide evidence that liraglutide could rescue tau hyperphosphorylation and the formation of Aβ plaque deposition which occurred under hyperinsulinemic and insulin-resistant conditions by resensitization of a neuronal insulin signaling as well as halting of overactivity of BACE-1, suggesting that liraglutide might exert a beneficial effect for AD subjects with an insulin-resistant condition.

## 4. Materials and Methods

### 4.1. Cell Culture

The human neuroblastoma cell line SH-SY5Y used in this study was obtained from American Type Culture Collection (ATCC No. CRL-2266) and maintained in DMEM/Ham F12 medium (Sigma-Aldrich, St. Louis, MO, USA), supplemented with 10% fetal bovine serum (FBS), 2 mM L-glutamine, 100 U/mL penicillin, and 100 µg/mL streptomycin in a 5% CO_2_ atmosphere at 37 °C. Cells at 80% confluence were subcultured every 3 days. For the experiments, cells were grown at a density of 1 × 10^5^ in 6 well plates which were approximately 50–60% confluence on the next day. The medium was changed every 24 h. The experimental protocol of the present study was shown in Figure 1.

### 4.2. Neuronal Insulin-Resistant Induction and Liraglutide Treatment

SH-SY5Y cells were incubated with high concentration (100 nM) of insulin for 2 days to induce neuronal insulin-resistant condition. The medium was changed every 24 h. Liraglutide (Victoza^®^, Novo Nordisk, Bagsvaerd, Denmark) 500 nM, either with or without insulin 100 nM, was added to cells on day 2 after inducing neuronal insulin resistance. For study protocol (as shown in Figure 1), those cells were used either to study insulin signaling, apoptotic protein expression, and Alzheimer’s markers. Experiment groups were identified as (1) control group (C), which was incubated in normal medium for 72 h; (2) liraglutide-treated group, which was incubated in normal medium for 48 h and then in medium containing 500 nM of liraglutide for further 24 h; (3) insulin resistant-group (I), which was incubated in medium containing 100 nM insulin for 72 h; and (4) liraglutide treated on insulin-resistant group (IL), in which 500 nM liraglutide was added for 24 h after the 48 h insulin-resistant induction. All groups were incubated in controlled temperature and humidity. For detecting the insulin signaling, cells were either left untreated or stimulated with 10 nM insulin for 30 min at 37 °C before harvesting. Cells were then washed with ice-cold phosphate buffer saline (PBS) and lysed with cell lysis buffer (20 mM Tris-HCl, 1 mM Na_3_VO_4_, a protease inhibitor cocktail (complete^TM^ Mini, 11836170001 Roche, Basel, Switzerland), and 5 mM NaF 1% NP40. Cell lysates were spun at 10,000× *g* for 30 min. Then, supernatants of cell lysates were collected and kept at −20 °C until used for immunoprecipitation and Western blot analysis.

### 4.3. Cell Viability Assay

Cell viability assay was adapted from [56]. Briefly, SH-SY5Y cells were seeded on a 96-well plate at a density of 1 × 10^4^ cells/well and incubated overnight with DMEM supplemented with 10% FBS. The medium was replaced by FBS-free medium, and cells were then treated with various concentrations of insulin (1–100 nM) or liraglutide (1–10^6^ nM) for 48 h and 24 h, respectively. Twenty microliters of 3-(4,5-dimethylthiazol-2-yl)-2,5-diphenyltetrazolium bromide, MTT (0.5 mg/mL in PBS) was added into the cells and incubated at 37 °C for 4 h. After incubation, the medium was removed and 200 μL of DMSO was added an incubated for 10 min in the dark. The absorbance at 590 nm was measured using a microplate reader (BioTek Instruments, Winooski, VT, USA). Cell viability was performed for three times, and each assay was done in triplicate.

### 4.4. Glucose Uptake Assay

Fluorescent glucose analog, 2-[N-(7-nitrobenz-2-oxa-1,3-diazol-4-yl) amino]-2-deoxy-d-glucose (2-NBDG) (Invitrogen, Carlsbad, CA, USA, Cat No: N13195) was used to measure glucose uptake into SH-SY5Y cells [57]. Cells were plated in 96-well plates at a density of 5 × 10^4^ cells/well and underwent the treatment as shown in Figure 1. On the experiment day (day3), cells were exposed to 80 mM 2-NBDG for 5 min in glucose-free DMEM. Cells were further washed with ice-cold HBSS containing Ca^2+^ and Mg^2+^ (Invitrogen, Cat No: 14025-092) to remove residual 2-NBDG. Fluorescent signal was detected using the microplate reader with excitation/emission of 465/540 nm.

### 4.5. Determination of Beta-Secretase (BACE-1) Activity

The BACE-1 activity was determined using a β-secretase (BACE-1) activity detection kit (Fluorescent) (Sigma-Aldrich, St. Louis, MO, USA). Cells were plated in 6-well plate at a density of 0.6 × 10^6^ cells/well and underwent the treatment as shown in Figure 1. On the experiment day (day 3), cells were lysed with cell lysis buffer (T-PER Tissue Protein Extraction Reagent, Thermo Scientific, Waltham, MA, USA). The amount of protein in each sample was ensured by NanoDrop^TM^ Lite spectophotometer, Thermo Scientific, USA). All reactions were performed using the 96-well microplate reader and monitored at excitation wavelength of 320 nm and emission wavelength of 405 nm as suggested by manufacturer’s recommendations. The activity of BACE-1 was reported as fold change compared to the control from three independent experiments.

### 4.6. Immunoprecipitation

Equality of the amount of protein in each sample was ensured by Bradford protein assay (Bio-Rad, Hercules, CA, USA). To isolate IRS-1 protein, cell lysates were immunoprecipitated overnight at 4 °C with 1 µg of IRS-1 antibody (Cell Signaling Technology, Inc, Danvers, MA, USA or Japan) and conjugated to protein A–agarose beads (company) for 24 h at 4 °C. After removing supernatant, bead complexes were washed three times with lysis buffer. The protein–bead complexes were eluted with 30 µL of 2× sample buffer (Glycerol 2 mL, SDS 6.0 g, Tris 1.4 g, with added double-distilled H_2_O (ddH_2_O) to 100 mL), boiled at 95 °C for 5 min, and spun at maximal speed for 1 min. Supernatants were subjected to Western blotting. An anti-Tyr antibody (Cell Signaling Technology, Inc) was used as a primary antibody to detect the phosphorylation status of IRS-1 protein.

### 4.7. Western Blot Analysis

After equilibrating the amount of protein, 2× sample buffer was added to each sample. The sample proteins were further separated by 10% sodium dodecyl sulfate polyacrylamide gel electrophoresis (SDS-PAGE) and then transferred to polyvinylidene fluoride (PVDF) membranes (Immobilon-P; Millipore, Bradford, MA, USA) for 1 h at 100 volts. The membranes were blocked with 5% nonfat milk in Tris-buffered Saline (TBS; Tris 24.2 g, NaCl 80 g, added ddH_2_O to 1000 mL, pH 7.4) containing 0.1% tween 20 for 1 h at 25 °C. The membranes were incubated with primary antibodies against phosphorylation of the insulin receptor Tyr 1162/1163, 1:1000 (Santa Cruz Biotechnology, Inc., Dallas, TX, USA), the insulin receptor β-subunit (1:1000, Santa Cruz Biotechnology, Inc., Dallas, TX, USA), phosphorylated AKT (Ser 473, 1:1000, Cell Signaling Technology, Inc, Danvers, MA, USA), AKT (1:1000, Cell Signaling Technology, Inc, Danvers, MA, USA), phosphorylated GSK-3β (Ser 9, 1:1000, Cell Signaling Technology, Inc, Danvers, MA, USA), GSK-3β (1:1000, Cell Signaling Technology, Inc, Danvers, MA, USA), Amyloid precursor protein (N-terminus, 1:1000, Merck Millipore, Burlington, MA, USA), Amyloid β (1:1000, Abcam, Cambridge, UK), phosphorylated tau (Ser 396, 1:1000, Invitrogen, Carlsbad, CA, USA), Tau5 (1:1000, Calbiochem, city, CA, USA), Bax (1:1000, Santa Cruz Biotechnology, Inc., Dallas, TX, USA), Bcl-2 (1:1000, Cell Signaling Technology, Inc, Danvers, MA, USA), and β-actin (1:1000, Santa Cruz Biotechnology, Inc., Dallas, TX, USA) at 4 °C overnight, followed by incubation with horseradish peroxidase-conjugated anti-rabbit IgG (1:2000, Santa Cruz Biotechnology, Inc., TX, USA). The protein bands were visualized by ECL Western blotting detection reagent and quantitated using Image J software.

### 4.8. Data Analysis

All data were expressed as mean ± standard error of mean (SEM). The Kruskal–Wallis Test, followed by Dunn’s post-hoc analysis, was used to determine the differences between groups. *p* < 0.05 was considered statistically significant. All experiments shown were performed with at least 3 individual experiments (*n* = 3).

## Figures and Tables

**Figure 1 ijms-21-01725-f001:**
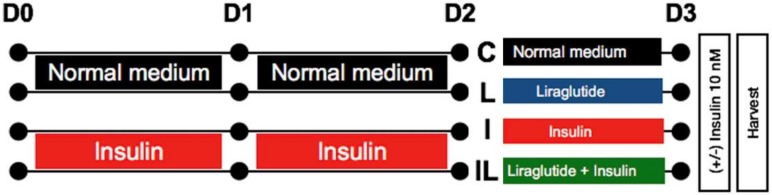
Schematic diagram of the experiment. Human neuroblostoma cells, SH-SY5Y, were incubated with normal medium or medium with 100 nM insulin for 48 h to induce insulin resistance. After 48 h, the cells without insulin induction were further treated with normal medium (C) or 500 nM liraglutide (L), while the insulin-resistant cells were further treated with 100 nM insulin (I) or 500 nM liraglutide (IL) for 24 h. Before harvesting cells, each group was subdivided into 2 groups (stimulated with 10 nM insulin or left unstimulated for 30 min). D0, D1, D2, and D3 represent the experimental days as day0, day1, day2, and day3, respectively, after cell seeding.

**Figure 2 ijms-21-01725-f002:**
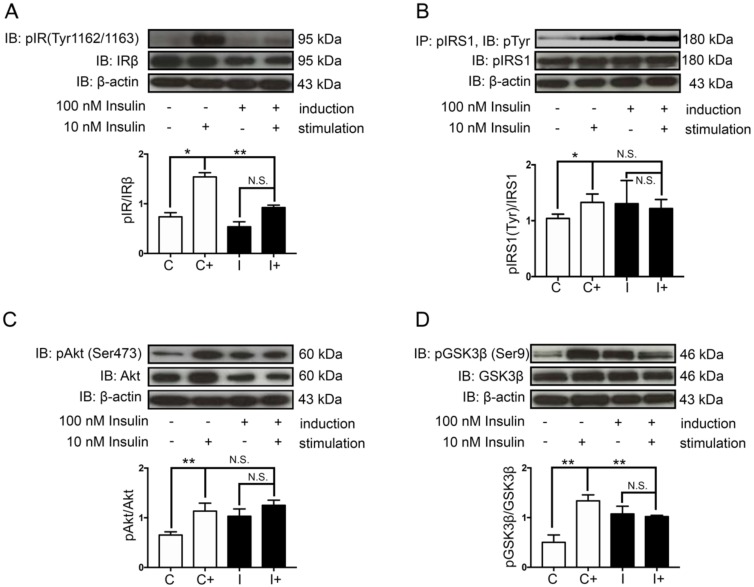
Effects of hyperinsulinemic conditions on insulin signaling pathway. The normal cells (C) and insulin-resistant cells (I) were subdivided and stimulated with 10 nM insulin (C+, I+) or without (C, I) for 30 min before harvesting. Western blotting bands and bar graphs represent densitometric values of each lane after normalizing with total protein expression; phosphorylated insulin receptor (pIR) (Tyr1162/1163)/IRβ (**A**) pAKT(Ser473)/AKT (**C**), pGSK-3β(Ser9)/GSK-3β (**D**). The bands and bar graphs of immunoprecipitated insulin receptor substrate 1 (IRS-1) followed by pTyr Western blotting of pIRS-1(Tyr)/IRS-1 representing densitometric values of each lane are shown in (**B**). Values are mean ± SEM, * *p* < 0.05, ** *p* < 0.01 compared with those of normal cells. All experiments shown were from *n* = 3.

**Figure 3 ijms-21-01725-f003:**
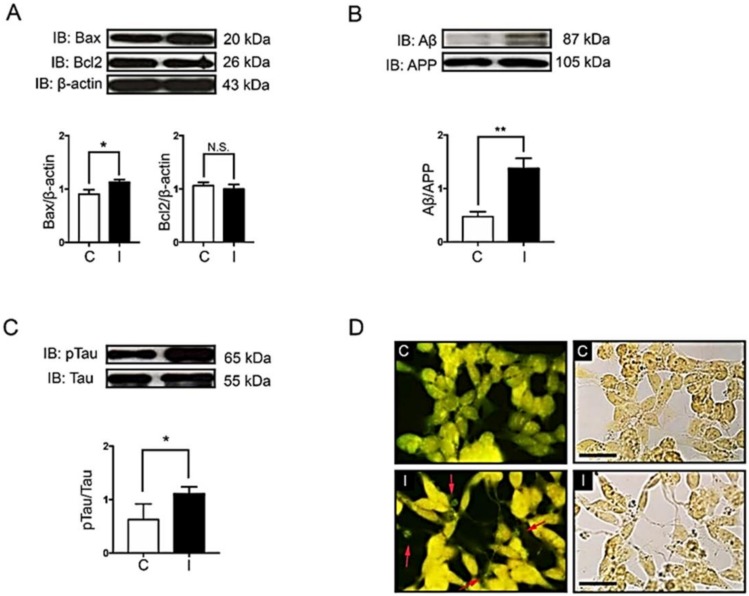
Effects of hyperinsulinemic conditions on apoptotic signals, Aβ and tau proteins. The normal cells (C) and insulin-resistant cells (I) were used to determine protein expression, including apoptotic signal proteins Bax and Bcl-2 (**A**), Aβ (**B**) and pTau (Ser396) (**C**). The Western blotting bands and bar graphs represent densitometric values of each lane after normalizing with total protein actin, amyloid precursor protein (APP), and tau protein, respectively. Values are mean ± SEM; * *p* < 0.05, ** *p* < 0.01 compared with normal cells (C). Thioflavin-S staining of neuritic plaques was done in SH-SY5Y cells (**D**). C and I represent the control and insulin-resistant groups, respectively. The stained cells were visualized by fluorescent microscope at 40X magnification. Red arrows indicate Thioflavin-S stained neuritic plaques. Scale bar, 25 µm. All experiments shown were from *n* = 3.

**Figure 4 ijms-21-01725-f004:**
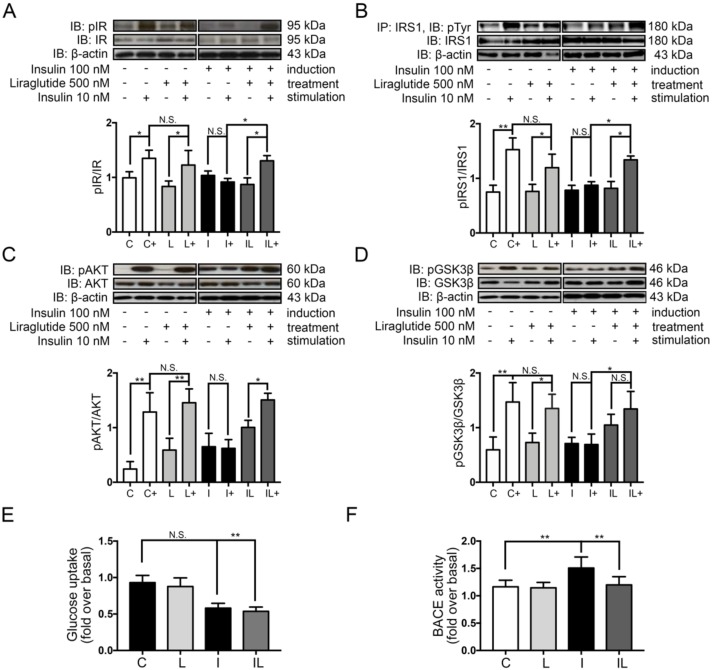
Effects of liraglutide on insulin signaling, glucose uptake, and BACE-1 activity. Western blots were performed for pIR(Tyr1162/1163), IR, pAKT(Ser473), AKT, pGSK-3β(Ser9), and GSK-3β. The bands and bar graphs represent densitometric values of each lane after normalizing with total protein expression; pIR(Tyr1162/1163)/IRβ (**A**), pAKT(Ser473)/AKT (**C**), pGSK-3β(Ser9)/GSK-3β (**D**). The bands and bar graphs of immunoprecipitated IRS-1 followed by pTyr Western blotting of pIRS-1(Tyr)/IRS-1 representing densitometric values of each lane are shown in (**B**). The effect of liraglutide treatment on glucose uptake is shown in (**E**)**.** The BACE-1 activity is shown in (**F**). Values are mean ± SEM, * *p* < 0.05 ** *p* < 0.01 compared with neuronal insulin-resistant cells. All experiments shown were from *n* = 3.

**Figure 5 ijms-21-01725-f005:**
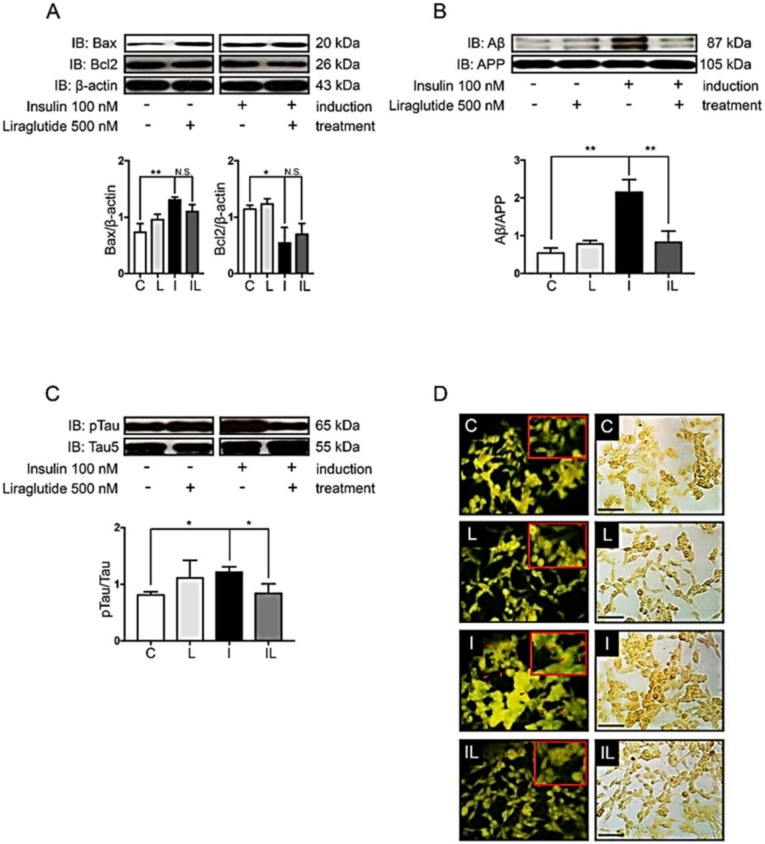
Effects of liraglutide on the formation of Alzheimer’s markers, plaque formation, and neuronal apoptosis. The normal cells (C) and insulin-resistant cells (I) insulin-resistant cells without liraglutide (L) or with liraglutide (IL) were used to determine protein expression including apoptotic signal proteins Bax and Bcl-2 (**A**), Aβ (**B**) and tau protein (**C**). The immunoreactive bands and bar graphs represent densitometric values of each lane after normalizing with total protein actin, amyloid precursor protein (APP), and tau protein, respectively. Values are mean ± SEM * *p* < 0.05, ** *p* < 0.01 compared with insulin-resistant cells. Thioflavin-S staining of neuritic plaques was done in SH-SY5Y cells (**D**). The normal cells without liraglutide (C) or with liraglutide (L), together with insulin-resistant cells without liraglutide (L) or with liraglutide (IL) treatment, were visualized by fluorescent microscope at 40× magnification. Red arrows indicate Thioflavin-S stained neuritic plaques. Scale bar 25 µm. All experiments shown were from *n* = 3.

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
