# Peer review of "Liraglutide Suppresses Tau Hyperphosphorylation, Amyloid Beta Accumulation through Regulating Neuronal Insulin Signaling and BACE-1 Activity"

_ijms, 2020, doi:10.3390/ijms21051725_

Round 1

Reviewer 1 Report

Authors tested liraglutide, a glucagon-like peptide-1 (GLP-1) analog, an
antidiabetic agent in SH-SY5Y cells and found that it could effectively reverse deleterious effects of insulin over-stimulation, by showing reversing the phosphorylation status of insulin receptors and its major downstream signaling molecules, and reducing the activity of BACE1. The research was carefully designed, experimental data are solid, and the interpretation is straightforward and sound. I have some minor point to be addressed before accepting for publication.

  1. In Abstract, check the grammar of the following sentence: 'Liraglutide, a glucagon-like peptide-1 (GLP-1) analog, an antidiabetic agent which has a benefit in improving a peripheral insulin resistance.' 
  2. line 11, 'over stimulation' -> over-stimulation
  3. line 14, 'enzyme which then affected the formation of beta-amyloid in insulin-resistant cells.- -> the meaning of word 'affected' is vague. Use other word, like 'decreased' or 'reduced'
  4. In Results, 2.1 Hyperinsulinemic condition induced neuronal insulin resistance in a human neuroblastoma cell line (SH-SY5Y) -> eliminate parenthesis.
  5. It would be helpful for readers to understan figures, if 'C, L, I, IL' are marked in the Fig. 1 diagram.
  6. In Fig.3 & 5 immunocytochemistry, show Thioflavin-S stained neuritic plaques as enlarged insets. The morphology and signals of the stained targets are not clear. One example of each stain should be shown in high resolution. Also, 'The black lines indicate 25 µm length.' is not a usual expression. 'Scale bar; 25 µm.' is common expression.
  7. P.13, line 9. 'liraglutide improves BACE1 activity which is accelerated under a hyperinsulinemic condition' -> The meaning of the word 'improve' is not clear. Use a word, like 'decrease' or 'suppress'.

Author Response

Response to reviwer1

  1. In Abstract, check the grammar of the following sentence: 'Liraglutide, a glucagon-like peptide-1 (GLP-1) analog, an antidiabetic agent which has a benefit in improving a peripheral insulin resistance.'

Response: We have already corrected the sentence (line 6, in abstract) to be a complete sentence.

  1. line 11, 'over stimulation' -> over-stimulation

Response: We have modified the sentence as suggested (line 11, Track Change)

  1. line 14, 'enzyme which then affected the formation of beta-amyloid in insulin-resistant cells.- -> the meaning of word 'affected' is vague. Use other word, like 'decreased' or 'reduced'

Response: We have changed “affected” to “ decreased” (line 14 Track Change)

  1. In Results, 2.1 Hyperinsulinemic condition induced neuronal insulin resistance in a human neuroblastoma cell line (SH-SY5Y) -> eliminate parenthesis.

Response: In Results 2.1, we eliminated parenthesis (line 20, Track Change).

  1. It would be helpful for readers to understan figures, if 'C, L, I, IL' are marked in the Fig. 1 diagram.

Response: We agree with the reviewer for this point. Therefore, we have mark the certain time point for 'C, L, I, IL' , and we added the new diagram.

  1. In Fig.3 & 5 immunocytochemistry, show Thioflavin-S stained neuritic plaques as enlarged insets. The morphology and signals of the stained targets are not clear. One example of each stain should be shown in high resolution. Also, 'The black lines indicate 25 µm length.' is not a usual expression. 'Scale bar; 25 µm.' is common expression.

Response:  Fig.3&5, we have changed new images to be clearer with high resolution.

We also replaced the old expression with “Scale bar, 25 µm.” as suggested.

  1. P.13, line 13. 'liraglutide improves BACE1 activity which is accelerated under a hyperinsulinemic condition' -> The meaning of the word 'improve' is not clear. Use a word, like 'decrease' or 'suppress'.

Response: We have changed the word 'improves' to “suppresses” as suggested.

Reviewer 2 Report

In this manuscript, the authors investigate the effect of the antidiabetic agent liraglutide in insulin-resistant-SH-SY5Y cells. The authors demonstrate the increase of Aβ, phosphorylated tau, plaque/tangles formation and BACE1 activity in hyperinsulinemic conditions, and their reduction and the improvement of insulin signaling after the treatment with liraglutide. Overall, this is an interesting study; the interconnection between AD and T2DM and the shared pathological mechanisms are accepted by the scientific community, and this manuscript proposes an antidiabetic agent as a possible anti-AD drug. I recommend publication after minor revisions listed below. 

  • In the introduction, the authors should briefly mention how insulin-signaling deficiencies are a contributing factor to tau hyperphosphorylation, neurofibrillary tangle formation and deposition of amyloid-β. There is a lot of literature in this regard. 
  • Despite all these prominent data, the restoration of glucose uptake was not observed. How do the authors explain that?
  • Cell cytotoxicity assay of SH-SY5Y cells with insulin induction and on 24-h liraglutide treatment should be added to the study.
  • We agree with the authors concerning the important role of metformin in paving the way to the development of new anti AD drugs starting from known treatments for diabetes. However, also silibyns, a natural product used to cure T2DM symptoms, has been recently demonstrated to have important anti-AD properties (see ACS chemical neuroscience 8 (8), 1767-1778, 2017). I think that the work on silibyins is worthy to be mentioned in the introduction. 

Author Response

Response reviewer2

1 In the introduction, the authors should briefly mention how insulin-signaling deficiencies are a contributing factor to tau hyperphosphorylation, neurofibrillary tangle formation and deposition of amyloid-β. There is a lot of literature in this regard.

Response: Thank you very much for suggestion. We have added the sentence and the references to mention the insulin signaling deficiency and its relevant consequences as labels in Track Change in the introduction section.

2 Despite all these prominent data, the restoration of glucose uptake was not observed. How do the authors explain that?

Response: The reason that glucose uptake was not restored after liraglutide treatment may possibly be due to incomplete rescue of insulin receptor (Fig.4A immunoblot lane 7-8). The observed restoration level may be enough to see the suppression of AD markers but may not reach the threshold for improving insulin-dependent glucose uptake. However, it is interesting to further explore in the future for the certain explanation for this phenomenon.

3 Cell cytotoxicity assay of SH-SY5Y cells with insulin induction and on 24-h liraglutide treatment should be added to the study.

Response: We did observe the behavior of cell for their viability at 24h treatment (insulin or liraglutide) and noticed similar trend of cell viability.

4 We agree with the authors concerning the important role of metformin in paving the way to the development of new anti AD drugs starting from known treatments for diabetes. However, also silibyns, a natural product used to cure T2DM symptoms, has been recently demonstrated to have important anti-AD properties (see ACS chemical neuroscience 8 (8), 1767-1778, 2017). I think that the work on silibyins is worthy to be mentioned in the introduction.

Response: We have added silybin study into the introduction as suggested.